# The Role of Different Lanthanoid and Transition Metals in Perovskite Gas Sensors

**DOI:** 10.3390/s21248462

**Published:** 2021-12-18

**Authors:** Abdulaziz Alharbi, Benjamin Junker, Mohammad Alduraibi, Ahmad Algarni, Udo Weimar, Nicolae Bârsan

**Affiliations:** 1National Center for Nanotechnology and Semiconductors, King Abdulaziz City for Science and Technology (KACST), Riyadh 11421, Saudi Arabia; abharbi@kacst.edu.sa (A.A.); agarni@kacst.edu.sa (A.A.); 2Institute of Physical and Theoretical Chemistry (IPTC), University of Tuebingen, Auf der Morgenstelle 15, D-72076 Tuebingen, Germany; benjamin.junker@ipc.uni-tuebingen.de (B.J.); upw@ipc.uni-tuebingen.de (U.W.); 3Center for Light-Matter Interaction, Sensors & Analytics (LISA+), University of Tuebingen, Auf der Morgenstelle 15, D-72076 Tuebingen, Germany; 4Physics and Astronomy Department, College of Science, King Saud University, Riyadh 11451, Saudi Arabia; malduraibi@ksu.edu.sa

**Keywords:** perovskites, gas sensor, DRIFTS, operando spectroscopy

## Abstract

Beginning with LaFeO_3_, a prominent perovskite-structured material used in the field of gas sensing, various perovskite-structured materials were prepared using sol–gel technique. The composition was systematically modified by replacing La with Sm and Gd, or Fe with Cr, Mn, Co, and Ni. The materials synthesized are comparable in grain size and morphology. DC resistance measurements performed on gas sensors reveal Fe-based compounds solely demonstrated effective sensing performance of acetylene and ethylene. Operando diffuse reflectance infrared Fourier transform spectroscopy shows the sensing mechanism is dependent on semiconductor properties of such materials, and that surface reactivity plays a key role in the sensing response. The replacement of A-site with various lanthanoid elements conserves surface reactivity of AFeO_3_, while changes at the B-site of LaBO_3_ lead to alterations in sensor surface chemistry.

## 1. Introduction

Calcium titanate (CaTiO_3_) was discovered in 1839 by German mineralogist Gustav Rose in the Ural Mountains (Russia). Perovskite material shares the same crystal structure as CaTiO_3_, and was subsequently named in honor of the Russian mineralogist Lev Alekseevich von Perovskiy [1]. Perovskites have played a potentially significant role over many decades in various breakthrough technologies as innovative functional materials [1,2,3]. For example, in 1986, Bednorz and Müller discovered high-temperature superconductivity (HTSC) based on perovskite material (cuprate ceramics). They were jointly awarded the Noble Prize in Physics in 1987 for their discovery [4]. Moreover, manganite, another perovskite-structured material, shows a colossal magneto-resistive (CMR) phenomenon, which is crucial in spintronic applications [5]. In recent years, perovskites are rapidly becoming promising materials for inexpensive and high-efficiency photovoltaic cells [6,7,8].

In the gas sensors field, perovskites are promising candidate materials in gas sensor applications, due to their unique electrical and catalytic properties [9,10,11,12,13,14,15]. Owing to the general formula of perovskite, written as ABO_3_, where the A-site cation bears a greater ionic radius than the B-site cation, this class of structures harbors a wide variety of possibilities for structure tailoring of oxides, i.e., by replacing A- and B-sites with different elements.

Among various target gases, the detection of hydrocarbons is extremely useful in a wide range of applications. For example, monitoring of dissolved gases (CH_4_, C_2_H_4_, C_2_H_2_, CO, CO_2_, and H_2_) in transformer oil provides beneficial information about transformer status [16,17]. Moreover, in agriculture, ethylene emissions indicate the maturity state of fruits; thus, detecting and controlling this gas is crucial for fruit ripening [18].

For the proper use and future development of selective gas sensors based on perovskite materials, it is essential to understand the sensing mechanism, including the molecular pathway of the reaction. However, few attempts were made to understand and systematically compare the underlying sensing mechanism of perovskites. For example, Arakawa and co-authors found a correlation between catalytic activity and the radius of A-site-element for LnFeO_3_, where Ln is a lanthanoid element. In the case of LnCrO_3_, however, the same correlation was unnoticeable. The effect of oxygen-binding energy with metal ions of perovskites on the gas-sensing mechanism was considered by the same group [19]. Furthermore, the influence of Ln elements and the surface composition of LnFeO_3_ on sensing NO_2_ was reported [20]. Subsequently, Siemons et al. used different lanthanoid elements (La, Pr, Nd, Sm, Eu, Gd, Tb, Dy, Ho, Er, Tm, Yb, and Lu) in LnFeO_3_ and LnCrO_3_ perovskite structures in order to investigate their gas-sensing properties of H_2_, CO, NO, NO_2_, and propylene via high-throughput impedance spectroscopy [21]. They noted a correlation between binding energy of oxygen to metal ions and gas sensing; the lower the binding energy, the greater the gas sensor signals (except for LuFeO_3_). Recently, Gaskov and co-workers reported that modification of LaCoO_3_ with Ag nanoparticles may lead to greater response and selectivity to H_2_S compared with pure LaCoO_3_. Additionally, they employed in situ infrared spectroscopy to study the chemical reactions of H_2_S on the sensor’s surface and observed an enhancement in H_2_S chemisorption as a result of Ag nanoparticles present [22]. A summary of relevant material properties is given in Table 1.

In our previous work, we used operando DRIFT spectroscopy to investigate the molecular pathway underlying the reaction of LaFeO_3_ (LFO)-based sensors during ethylene and acetylene exposure. We demonstrated that the sensor response of LFO is associated with formation of surface formats as opposed to oxidation–reduction of the oxide surface [40].

This work aims to examine the validity of our novel mechanism for different perovskite compounds where La is replaced by other trivalent lanthanoid ions (Sm, Gd) or the transition metal ion Fe at the B-site is replaced by Cr, Mn, Ni, and Co. This systematic approach allows us to assess the role of both metals in the gas-sensing process. The sensing properties of all perovskite materials towards various hydrocarbons were examined. Moreover, operando DRIFT spectroscopy was used to investigate the molecular interaction between hydrocarbons and perovskite surfaces.

## 2. Experimental Details

### 2.1. Synthesis and Structural Characterization of Perovskites

Sol–gel technique was used to prepare LaCrO_3_, LaNiO_3_, LaCoO_3_, LaMnO_3_, LaFeO_3_, SmFeO_3_, and GdFeO_3_ perovskite powders. Stoichiometric amounts of different metal nitrates were used as received, shown in Table 2, and dissolved using a (1:1) ratio of citric acid for each compound. Each mixture was then dissolved in deionized water. Following this, each solution was neutralized by adding ammonium hydroxide into the vigorously stirred precursor solution. The obtained gel was dried at 90 °C for 4 h. Finally, the powder was calcined at 600 °C for 2 h.

To confirm the perovskite phase formation of the prepared powders, X-ray diffractometers (MiniFlex 600, Rigaku, Tokyo, Japan and D8 discover, Bruker, Billerica, MA, USA) were used with nickel filtered Cu-Kα and Co-Kα radiation, respectively, in the diffraction range of 0° to 65°. Moreover, the morphologies of the perovskite powders were investigated using a field emission scanning electron microscope (FE-SEM, JEOL-7600F, Tokyo, Japan).

### 2.2. DC Measurements

Thick film sensors were prepared by screen printing the powders onto alumina substrates as described in [41]. Afterwards, the sensors were calcined at 500 °C to remove residual organic solvents in the paste. For DC resistance measurements, the sensors were mounted in a test chamber connected to a gas-mixing system. The resistive heater on the backside of each sensor substrate was individually calibrated with an infrared thermometer (KTR2300, Maurer, Kohlberg, Germany). The sensors were mounted in a PTFE measurement chamber and exposed to test gases in a background of 0%, 10%, and 50% relative humidity (measured at 25 °C) at operating temperatures of 150 °C and 250 °C. The sensors were exposed to concentrations of 50, 100, 200, and 500 ppm of methane, ethane, ethylene, and acetylene for 30 min. Analyte gases (Westfalen AG) were mixed with dry and humid air in a gas-mixing system equipped with mass flow controllers (Bronkhorst). The resistance was measured with a Keysight 34972A multimeter. The range of the device was increased as necessary by connecting a 100MOhm precision resistor in parallel to the sensor.

### 2.3. Operando DRIFT Spectroscopy

DRIFTS was performed on a Bruker Vertex 80 v spectrometer equipped with DLaTGS and MCT detectors. Absorbance spectra were calculated from the spectra which were recorded after two hours of the gas exposure referenced to the ones which were taken initially in clean air. DRIFTS experiments were performed on sensors as described above at 150 °C in dry air.

## 3. Results

### 3.1. Material Characterization

Figure 1 shows the XRD patterns of synthesized LaCrO_3_, LaNiO_3_, LaCoO_3_ LaMnO_3_, LaFeO_3_, SmFeO_3_, and GdFeO_3_ materials. Due to our usage of two different X-ray sources, one with Cu and one with Co anode, we separated our XRD results into two figures, Figure 1a,b. Moreover, each figure contains its reference peaks, card no. JCPDS 37-1493 in Figure 1a and card no. ICSD 204,689 in Figure 1b. All samples show peaks related to perovskites structure.

The SEM images of the prepared perovskites are shown in Figure 2. All samples appear to contain nanoparticles with shapes of similar uniformity. The grain sizes range primarily between 50–100 nm diameter. XRD results confirm SEM findings indicating the following grain sizes: LaFeO_3_: 63 nm, LaMnO_3_: 61 nm, LaCoO_3_: 80 nm, LaNiO_3_: 38 nm, LaCrO_3_: 42 nm, GdFeO_3_: 92 nm, and SmFeO_3_: 101 nm. Fe-based compounds show less nanoparticle agglomeration and tend to form mesopores with greater ease than other compounds.

### 3.2. Electrical Characterization

To determine whether the materials react as p-type or n-type semiconductors, sensors were equilibrated in dry nitrogen and subsequently exposed to dry air. The change in resistance is plotted in Figure 3a. The sensor signals (defined as $\frac{R_{nitrogen}}{R_{air}}$) range from 1.1 (LaCoO_3_) to 89 (GdFeO_3_). For all materials, a decrease in resistance was observed, except for LaMnO_3_, where resistance increased (S = 0.98). This indicates all materials except LaMnO_3_ react as p-type semiconductors. The results are supported by at least one of the references given in Table 1.

Figure 3b shows the resistance of gas sensors during exposure to different analyte gases in dry air at 150 °C. When La ions are replaced by Sm, the baseline resistance increases by a factor of 10, while replacing La with Gd increases the baseline resistance by another order of magnitude. When the Fe site is replaced by other transition metal ions, the opposite behavior is observed. For LaCrO_3_ and LaMnO_3_ the resistance is significantly decreased. For LaCoO_3_ and LaNiO_3_, the resistance ranges from 10 to 20 Ohms, which is comparable to the resistance of the metallic electrodes on the substrates.

The response to the analyte gases tested in this study varies substantially between Fe-based compounds and their counterparts. The response to saturated hydrocarbons methane and ethane is small for all materials. In contrast, ethylene and acetylene are clearly detected by all materials containing Fe, while the response is far lower for other materials. LaMnO_3_ is the only material that shows a decrease in resistance upon contact with reducing gases. This is in keeping with findings of the oxygen exposure results. Interestingly, in the cases of LaFeO_3_, SmFeO_3_, and GdFeO_3_, 30 min for one step of concentration was sufficient to reach equilibrium for acetylene, but not ethylene, even though the change in resistance is comparable. The reaction with acetylene is apparently faster at this temperature. At 250 °C (Figure 3b) the signals are smaller and the response is faster for all material–target gas combinations. For ethylene, this effect is more pronounced than acetylene. The baseline resistance is decreased for all materials except LaNiO_3_. The low resistance in addition to the negative temperature coefficient of resistance represent properties of a metal rather than a semiconductor.

Figure 4 summarizes sensor signals for all investigated materials in different conditions. Methane remained undetected except for low signals from GdFeO_3_ and SmFeO_3_ sensors at 150 °C. Regarding ethane, the situation is similar, however the gas is detectable with the LaFeO_3_ sensor and with greater sensor signals than methane. At 250 °C, a small sensor response may be observed with Fe-containing materials. For ethylene, sensor signals are greater by several orders of magnitude for Fe-containing materials. In a humid background, the sensor signals are smaller than in dry air. The sensor response to acetylene is more influenced by humidity than ethylene. This is due to dissociation of acetylene at the sensor surface being inhibited by the additional formation of OH groups, as demonstrated in our previous work [40]. The agreement of this study’s data with earlier publications confirms reproducibility of the results [9]. Among other materials, only LaCrO_3_ shows a response undiminished in a humid background. At greater temperatures the signals for ethylene are significantly decreased, particularly in humid air where only GdFeO_3_ shows a response. For acetylene, the influence of humidity on materials containing Fe is more pronounced than for ethylene. With other materials showing a response, i.e., LaCrO_3_, LaCoO_3_, and LaNiO_3_, the influence of humidity remains small.

These findings indicate Fe ions at the B-site of this class of materials play a key role in the gas-sensing mechanism, and that the occurrence of different surface reactions are expected, as observed from the different response of materials to acetylene and ethylene. The sensor response increases with baseline resistance of Fe-containing materials. The sensor signals of LaNiO_3_ and LaCoO_3_, however, reveal that a low baseline resistance does not necessarily exclude a response. A potential explanation for the differences among the sensor signals of Fe-containing materials may be their different morphology.

### 3.3. Surface Characterization

To compare the surface reactions on various materials, operando DRIFT spectroscopy was applied. The condition of 500 ppm of acetylene in dry air at 150 °C was chosen for comparison, as a majority of materials show a sensor signal. Additionally, the detection of acetylene is less influenced by the operating temperature and electrode material. The absorbance spectra with dry air as reference are presented in Figure 5. The presence of two bands at 2959 and 2850 cm^−1^, together with 1580 and 1377 cm^−1^, indicates the formation of formate species [40,42]. The two peaks at 2959 and 2850 cm^−1^ are attributed to C-H vibrations (asymmetric and symmetric), and the 1580 and 1377 cm^−1^ bands are associated with OCO vibrations (asymmetric and symmetric). The peak at 1430 cm^−1^ may be assigned to δ C-H [43,44]. Moreover, two small peaks in the OH region can be observed: a decreasing peak in around 3671 cm^−1^ indicates consumption of terminal OH groups.

These peaks are additionally present on GdFeO_3_ and SmFeO_3_, but with slightly different relative heights. Notably, the ratios of peaks in the OH region differ, indicating either a marginally different reaction or a different surface coverage prior to acetylene exposure. For GdFeO_3_, gaseous CO_2_ is visible around 2351 cm^−1^. The situation concerning materials LaMnO_3_, LaCoO_3_, and LaNiO_3_ is very different. Change at the surface cannot be observed. These materials exhibit low reflectance and the low intensity at the detector leads to high noise levels. A very different behavior is observed for LaCrO_3_. The same formate peaks as seen with LaFeO_3_ are observed, albeit at a much lower intensity. A broad band around 3400 cm^−1^ indicates the formation of interacting hydroxyl groups. Moreover, the distinctive double band for gaseous CO_2_ can clearly be seen at 2351 cm^−1^. Based on our previous investigation [40], complete combustion of hydrocarbons, indicated by CO_2_ formation in DRIFT spectra, is not generally accompanied by a strong sensor response. However, incomplete combustion of hydrocarbons, which can lead to the formation of formate species at the perovskite surface, playing a key role in effective gas sensing. As observable from DRIFT spectra, the complete combustion section of surface chemical reactions is more pronounced for LaCrO3 than Fe-based materials; therefore, the LaCrO3 sensors show a lower gas response.

Kremenic et al. investigated the catalytic properties of a series of LaBO_3_ (B=Cr, Mn, Fe; Co, Ni) perovskites. In this series, LaFeO_3_ bears the greatest tendency for absorption of isobutene, but the lowest for oxygen [45]. Moreover, the catalytic conversion of isobutene at temperatures comparable to our study was investigated. The lowest reaction rate for complete combustions for LaFeO_3_ was observed, followed by LaCrO_3_. The incomplete combustion was assessed by analyzing the amount of oxygenated organic compounds where the highest values were found for LaCrO_3_ and LaFeO_3_. These results support the assumption that the incomplete combustion of hydrocarbons and strong sensor performance is correlated.

The molecular orbitals (MO) with symmetry e_g_ centered at the B-site of the perovskite with the highest energy and e_g_ symmetry are populated by 0 (Cr^3+^), 1 (Mn^3+^, Co^3+^, Ni^3+^), or 2 (Fe^3+^) electrons and interact differently with simple gas molecules such as CO, NO or O_2_ [46]. The reaction sequence involves multiple intermediates, whose energies are dependent on the composition of the perovskite [47] The creation of formates requires preadsorbed oxygen, provided by oxygen in the gas phase. This is consistent with the finding that materials with greater signals to oxygen additionally show a better response to hydrocarbons. Moreover, we assume the reception of gas molecules investigated in this study relies on the transfer of electrons between molecular orbitals of the hydrocarbons and the B-site cations of the perovskite. The structure of the analyte gases used in this study is more complex than O_2_ or CO. In contrast to methane and ethane, acetylene and ethylene contain additional π-MOs to interact with metals. A possible explanation for the high sensor signals of ethylene and acetylene may be interaction of these π-MOs with d*_x_*^2^_−*y*_^2^ MO of the Fe centers. This orbital is occupied solely for LnFeO_3_ [46]. This effect in combination with different reactivity to oxygen is suspected to be responsible for different sensing performances of materials with different B-sites. The exchange of La with Sm or Gd does not alter the number of electrons in MOs at the B-site. However, other factors such as ionic radii, electronegativity, magnetic properties, and differences in morphology of sensing layers may explain different sensor responses. Further research on surface chemistry and DFT calculations is planned to investigate the interplay between the perovskite surface, oxygen, and hydrocarbon molecules.

## 4. Conclusions

Lanthanoid-based perovskites present a promising class of materials for gas sensing due to their notable response to unsaturated hydrocarbons at low temperatures. As expected from the chemical properties of pure elements, exchanging different transition metal cations at the B-site drastically alters the electrical, chemical, and optical properties of the perovskite. Only Fe-based compounds showed clear sensor signals to acetylene and ethylene among other prepared perovskites. It appears from DRIFT spectra that surface reactivity, particularly the formation of formates, determines sensing behavior. In the case of LaCrO_3_, which additionally showed clear surface reactivity, the lower response to acetylene may be attributed to complete combustion of gases. Therefore, the most promising element for gas-sensing applications at the B-site is iron. In addition, modifications on the A-site (lanthanoids) of Fe-based perovskites maintain overall gas-sensing properties.

## Figures and Tables

**Figure 1 sensors-21-08462-f001:**
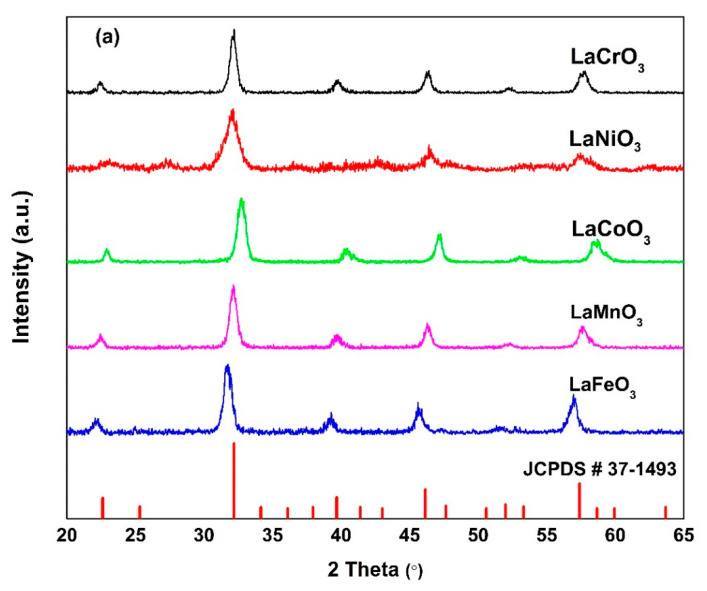
The XRD patterns of prepared perovskite materials using (**a**) Cu anode and (**b**) Co anode. The referenced peaks are indicated by red lines at the bottom.

**Figure 2 sensors-21-08462-f002:**
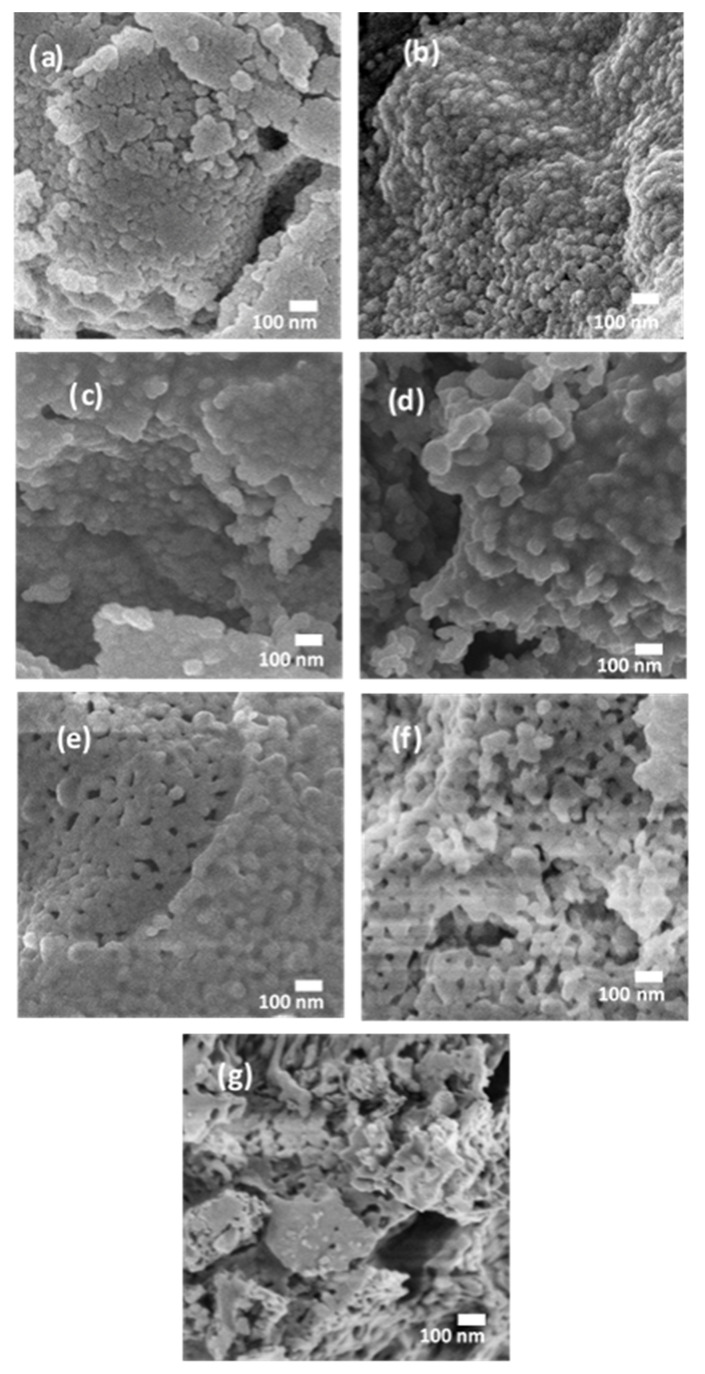
(**a**) LaCrO_3_, (**b**) LaNiO_3_, (**c**) LaCoO_3_, (**d**) LaMnO_3_, (**e**) LaFeO_3_, (**f**) SmFeO_3_ and (**g**) GdFeO_3_.

**Figure 3 sensors-21-08462-f003:**
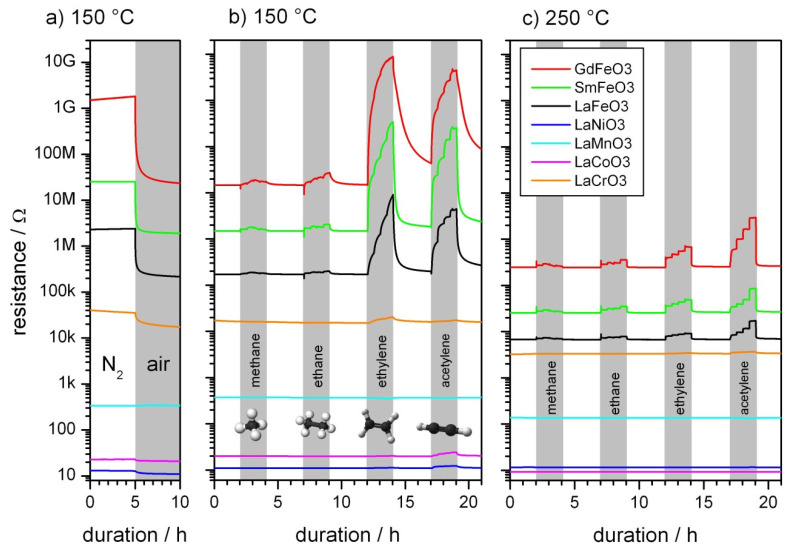
DC resistance measurements in dry air: (**a**) transition from nitrogen to air at 150 °C, (**b**) exposure different analyte gases at 150 °C, and (**c**) 250 °C. The shaded areas indicate periods, where the sensors were exposed to different concentrations of the analyte gases.

**Figure 4 sensors-21-08462-f004:**
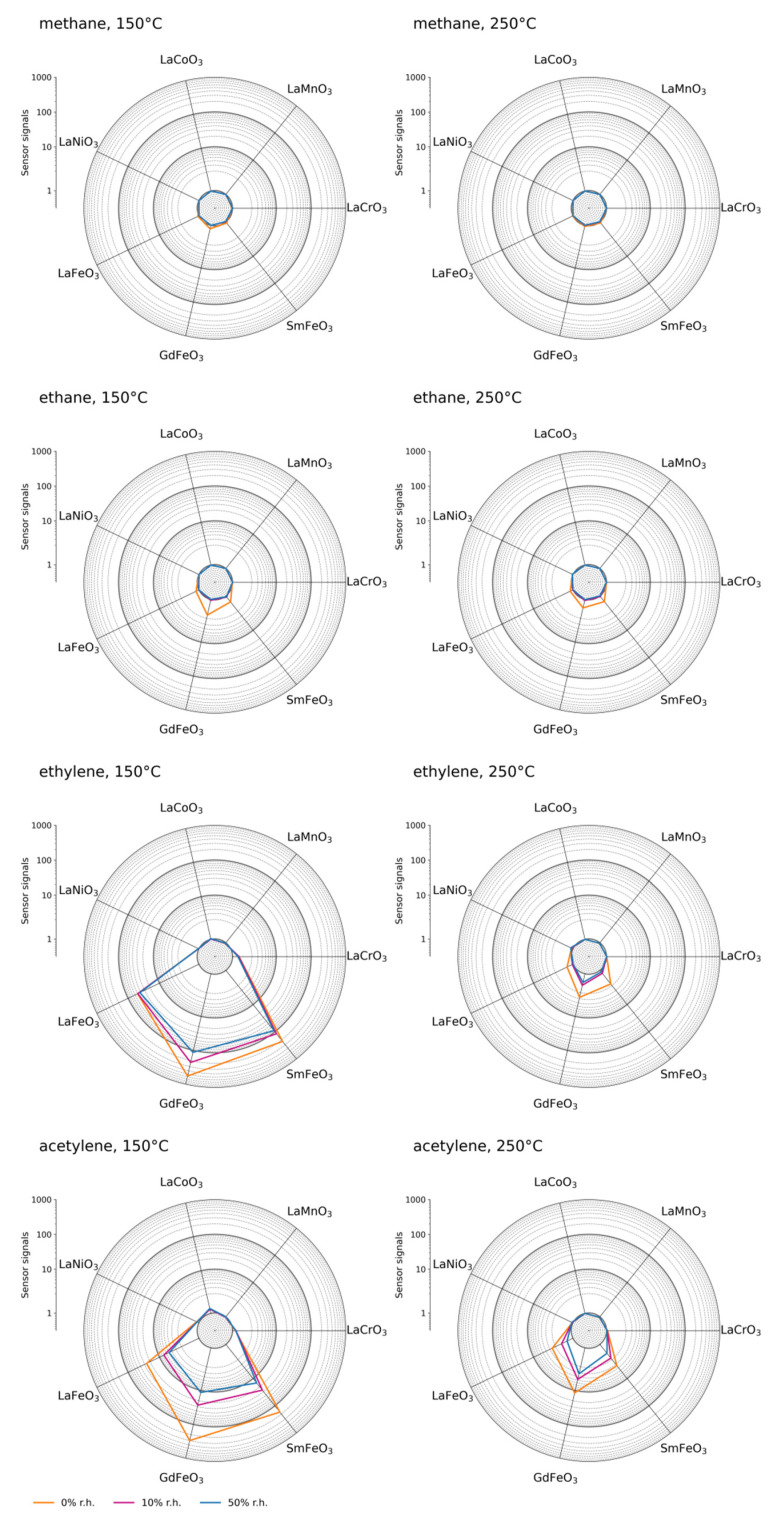
Sensor signals for exposure to 500 ppm of different gases at 150 °C (**left column**) and at 250 °C (**right column**).

**Figure 5 sensors-21-08462-f005:**
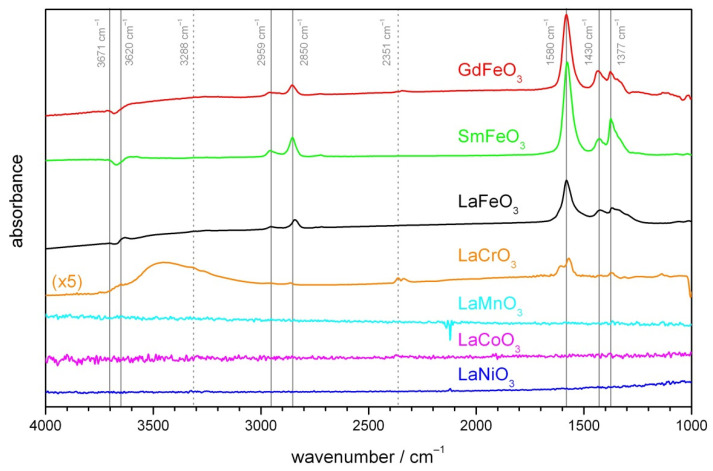
Absorbance DRIFT spectra of different materials at 150 °C. The spectra recorded after 2 h of exposure to 500 ppm acetylene were referenced to dry air. For increased visualization, the spectra are stacked and the magnitude of the spectrum of LaCrO_3_ increases. The dashed lines mark the position of gas phase species.

**Table 1 sensors-21-08462-t001:** Properties of different perovskite materials.

Material	Bandgap	Conduction Type	Color
LaCrO_3_	3.4 eV [23,24] 2.8 eV [25]	p-type [21] n-type [19]	dark green
LaNiO_3_	metallic [23,26]	metallic p-type [27,28]	black
LaCoO_3_	2.2 eV [29,30] 0.6 eV [23,31]	p-type [22] n-type [19]	black
LaMnO_3_	0.7 eV [23,32] 0.33 eV [33]	p-type [34] n-type [19]	black
LaFeO_3_	2.6 eV [35] 2.1 eV [23,36]	p-type [19,21]	light brown
SmFeO_3_	2.2 eV [37]	p-type [21]	light brown
GdFeO_3_	3.5 eV [38] 2.3 eV [39]	p-type [21]	brown

**Table 2 sensors-21-08462-t002:** Details of the raw materials used for sensitive materials preparation.

Perovskite Material	Metal Precursors
LaCrO_3_	La(NO_3_)_3_ · 6H_2_O (Fluka, (Buchs, Switzerland), Puriss. p.a., ≥99.0%) and Cr(NO_3_)_2_.9H_2_O (Sigma, (Buchs, Switzerland), Puriss. p.a., ≥99.0%)
LaNiO_3_	La(NO_3_)_3_ · 6H_2_O (Fluka, (Buchs, Switzerland), Puriss. p.a., ≥99.0%) and Ni(NO_3_)_2_.6H_2_O (Sigma, (Buchs, Switzerland), Puriss. p.a., ≥99.0%)
LaCoO_3_	La(NO_3_)_3_ · 6H_2_O (Fluka, (Buchs, Switzerland), Puriss. p.a., ≥99.0%) and Co(NO_3_)_2_ · 6H_2_O (Sigma, (Buchs, Switzerland), Puriss. p.a., ≥99.0%)
LaMnO_3_	La(NO_3_)_3_ · 6H_2_O (Fluka, (Buchs, Switzerland), Puriss. p.a., ≥99.0%) and Mn(NO_3_)_2_ · 4H_2_O (Sigma, (Buchs, Switzerland), Puriss. p.a., ≥99.0%)
LaFeO_3_	La(NO_3_)_3_ · 6H_2_O (Fluka, (Buchs, Switzerland), Puriss. p.a., ≥99.0%) and Fe(NO_3_)_3_ · 9H_2_O (Fluka, (Buchs, Switzerland), Puriss. p.a., ≥99.0%)
SmFeO_3_	Sm(NO_3_)_3_ · 6H_2_O (Acros Organics, (Geel, Belgium), ≥99.9%) and Fe(NO_3_)_3_ · 9H_2_O (Sigma Aldrich, (Buchs, Switzerland), ≥99.0%)
GdFeO_3_	Gd(NO_3_)_3_ · 6H_2_O (Aldrich, ≥99.9%) and Fe(NO_3_)_3_ · 9H_2_O (Sigma Aldrich, (Buchs, Switzerland), ≥99.0%)

## Data Availability

The data presented in this study are available on request from the corresponding author.

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
