# Peer review of "The Role of Different Lanthanoid and Transition Metals in Perovskite Gas Sensors"

_sensors, 2021, doi:10.3390/s21248462_

Round 1
Reviewer 1 Report
This paper reports the study of the acetylene gas sensing mechanism of ABO3 perovskites employing operando DRIFTS. The results are of enough value to deserve publication in Sensors. However, there are a few minor aspects that need improvement. These are as follows:
1. The choice of acetylene only (ethylene is discarded) for conducting the DRIFTS study needs to be better explained.
2. Figure 4 needs to be improved. The font size is too small to enable reading. Since there are no error bars associated to the values shown in these radar plots, it is impossible to determine whether or not the slight differences in ethylene and acethylene responses reported for AFeO3 perovskites are significant.
3. Can you give a reason for acethylene response being more affected by background moisture than the response to ethylene? This deserves a short discussion.
4. It is indicated that a different mechanism involving the combustion of acethylene holds for LaCrO3. However, the DC response for this perovskite when exposed to acethylene is very poor. Maybe this aspect should be better discussed.
5. Finally, the conclusions are too short and do not involve all the different findings presented in the paper. They should be improved as well.
Author Response
Reviewer 1
Dear Reviewer, thanks a lot for your observations. Please find below our answers, inserted after each of your question:
This paper reports the study of the acetylene gas sensing mechanism of ABO3 perovskites employing operando DRIFTS. The results are of enough value to deserve publication in Sensors. However, there are a few minor aspects that need improvement. These are as follows:
- The choice of acetylene only (ethylene is discarded) for conducting the DRIFTS study needs to be better explained.
- Both acetylene and ethylene are suitable candidates for a comparison of the different materials. In our previous works we could show, that both of them lead to the formation of formate species at temperatures where the sensor signals are highest (150°C). A difference in between the materials however is, that the sensor signal of ethylene is strongly dependent on the nature of the electrodes (Au or Pt). For acetylene we expect a simpler chemistry. The following paragraph was added: “The condition of 500 ppm of acetylene in dry air at 150 °C was chosen for the comparison, because most of the materials show at least some sensor signal and because the detection of acetylene is less influenced by the operating temperature and the electrode material.”
- Figure 4 needs to be improved. The font size is too small to enable reading. Since there are no error bars associated to the values shown in these radar plots, it is impossible to determine whether or not the slight differences in ethylene and acethylene responses reported for AFeO3 perovskites are significant.
- We agree that the font size in this Figure should be increased and improved this in the updated manuscript. For acetylene and ethylene we added a reference to the manuscript where the results for LaFeO3 have already been published in a peer-reviewed journal. For the other materials, we conducted multiple experiments as well, which show that the differences are significant.
- Can you give a reason for acetylene response being more affected by background moisture than the response to ethylene? This deserves a short discussion.
- Thanks for your comment. We added the following brief discussion in the revised manuscript: “The sensor response to acetylene is more influenced by humidity than ethylene. This is because the dissociation of acetylene at the sensor surface is inhibited by the additional formation of OH groups as we demonstrated in our previous work” (see reference 23 in the manuscript).
- It is indicated that a different mechanism involving the combustion of acethylene holds for LaCrO3. However, the DC response for this perovskite when exposed to acethylene is very poor. Maybe this aspect should be better discussed.
- We added the following brief discussion in the revised manuscript: “Based on our previous investigation (see reference 23 in the manuscript), complete combustion of hydrocarbons – indicated by CO2 formation in DRIFT spectra – is not generally accompanied by a good sensor response. However, incomplete combustion of hydrocarbons, which can lead to the formation of formate species at perovskite surface, plays a key role in good gas sensing. As it can be seen from DRIFT spectra, the complete combustion part of the surface chemical reactions is more pronounced for LaCrO3 than for Fe based materials; therefore, the LaCrO3 sensors show less gas response.”
- Finally, the conclusions are too short and do not involve all the different findings presented in the paper. They should be improved as well.
- The paragraph with conclusions has been completely revised and was extended.

Reviewer 2 Report
The paper describes interesting results, however, the conclusions made by authors require some additional evidence and the presentation of results must be improved.
- The sensing experiment is not described at all – what were the concentrations of the gases, how were they generated, what was the sensor chamber, what was the gas flow protocol. Must be provided.
- The obtained materials are not sufficiently characterized. Concerning the database cards – name the substances to which they correspond. Explain the shift in the reflection positions, observed on fig 1a. Explain the absence of many reflections from JCPDS #37-1493 card on the diffractograms, while on the figure 1b all reflections are present. Can the grain size be calculated from XRD patterns? Can the content of the amorphous phase be estimated?
- The materials effective surface area must be measured.
- Table 3 is better to support or replace with the figure of resistance change.
- 4 is not informative at all. What is the reason of putting the materials on it in such particular clockwise order? I suppose that the figure should be replaced with simple 2D plot.
- What concentrations were used for DRIFT spectroscopy? Why 2h period was chosen to collect the spectra of adsorbed species? Maybe the shorter times will allow to detect the absorbance bands for Ni, Co and Mn containing perovskites as well?
- The table 3 and fig. 3 indicate that the same materials, which have the strongest response to oxygen, have the strongest response to hydrocarbons. Can it be connected?
- The paragraph between lines 252 and 260 must be rewritten – the DRIFT spectra show you only HCO2- groups, not ethylene or acetylene molecule adsorption. Thus, addressing the observed sensor response to pi-MO system only is incorrect. The oxygen should play role as well. Maybe thermo-programmed reduction of obtained materials or thermo-programmed desorption of probe molecules can give rise to the supported ideas on the origins of sensor effect.
- Lines 256-258 “In contrast to methane and ethane, acetylene and ethylene have π-MOs to interact with the metals. Their interaction with Fe-centres, where the dx2-y2 MO is occupied may be the key to the gas sensing mechanism [46].” This notion is vaguely supported by the provided reference. The authors should provide extended explanations.
- Lines 72-73 “We demonstrated that the sensor response of LFO is associated with the 72 formation of surface formates, not the formation/cancellation of surface vacancies.” – there are no investigation methods, reported in the manuscript, which deal with the oxygen vacancies. The sentence is unsupported.
- Lines 79-80 “Moreover, operando DRIFT Spectroscopy was used to investigate the sensing 79 ” – talking about a molecular pathway of the reaction would be more correct, than using the term “sensing mechanism”.
Author Response
Reviewer 2
Dear Reviewer, thanks a lot for your observations. Please find below our answers, inserted after each of your question:
The paper describes interesting results, however, the conclusions made by authors require some additional evidence and the presentation of results must be improved.
- The sensing experiment is not described at all – what were the concentrations of the gases, how were they generated, what was the sensor chamber, what was the gas flow protocol. Must be provided.
- The required information has been provided in more detail in the experimental section: “Thick film sensors have been prepared by screen printing the powders onto alumina substrates as described in [41]. Afterwards the sensors have been calcined at 500 °C to remove residuals of organic solvents in the paste. For DC resistance measurements the sensors have been mounted in a test chamber connected to a gas mixing system. The resistive heater on the backside of each sensor substrate was individually calibrated with an infrared thermometer (Maurer KTR2300). The sensors were mounted in a PTFE measurement chamber and were exposed to test gases in in a background of 0%, 10%, and 50% relative humidity (measured at 25 °C) at operating temperatures of 150 °C and 250 °C. The sensors were exposed to concentrations of 50, 100, 200, and 500 ppm of methane, ethane, ethylene, and acetylene with a time of 30°min for each concentration. Analyte gases (Westfalen AG) were mixed with dry and humid air in a gas mixing system equipped with mass flow controllers (Bronkhorst). The resistance was measured with a Keysight 34972A multimeter. Where necessary the range of the device has been increased by connecting a 100MOhm precision resistor in parallel to the sensor.”
- The obtained materials are not sufficiently characterized. Concerning the database cards – name the substances to which they correspond. Explain the shift in the reflection positions, observed on fig 1a. Explain the absence of many reflections from JCPDS #37-1493 card on the diffractograms, while on the figure 1b all reflections are present. Can the grain size be calculated from XRD patterns? Can the content of the amorphous phase be estimated?
- JCPDS #37-1493 card is related to LaFeO3 which we used as a reference to confirm the presence of perovskite phase. The absence of small peaks is due to the noise, but, for each sample, the main peaks of perovskite are clearly observed. According to the XRD results, the grain size of LaFeO3, LaMnO3, LaCoO3, LaNiO3, LaCrO3, GdFeO3 and SmFeO3 are 63, 61, 80, 38, 42, 92 and 101 nm, respectively, which is in line with what we obtained from the SEM. We provided this information in the manuscript.
- The materials effective surface area must be measured.
- We respectively disagree with the reviewer because the SEM results don’t indicate major differences in porosity and the differences in sensor signals between the materials cannot be attributed to different specific surfaces. Anyways, it will take us a lot of time to get these results because this technique is not available in house. Accordingly we are asking the reviewer to abandon his demand for this investigation.
- Table 3 is better to support or replace with the figure of resistance change.
- We decided to add an additional panel to Figure 3 where the change of resistance is shown five hours before and after the switch to air. The table was removed from the manuscript and the most relevant numbers are included in the text.
- 4 is not informative at all. What is the reason of putting the materials on it in such particular clockwise order? I suppose that the figure should be replaced with simple 2D plot.
- In our opinion, radar plots are an effective way to convey complex data in a concise and informative way, so that the differences can easily spotted at one glance. The data we present have four dimensions (temperature, gas type, material and background humidity) and would at least require four two-dimensional plots as well. The sensor signals are close to one in many conditions and comparing these to some of the highest ones (>100) will be challenging in any type of plot. However we adjusted the font sizes and labels of the plot to make it more straightforward.
- What concentrations were used for DRIFT spectroscopy?
- As specified twice in the manuscript (at the beginning of the section “surface characterization” and caption of Figure 5), the concentration of 500 ppm acetylene was used for DRIFTS.
- Why 2h period was chosen to collect the spectra of adsorbed species? Maybe the shorter times will allow to detect the absorbance bands for Ni, Co and Mn containing perovskites as well?
- During the two hour exposures to the test gas we recorded DRIFTS spectra each 15 min, which is the time needed to acquire a high quality spectrum. We did not observe any qualitative difference between the eight spectra acquired during one exposure. In the manuscript we present the results at the end to make sure they are representative for equilibrium sensor signal.
- The table 3 and fig. 3 indicate that the same materials, which have the strongest response to oxygen, have the strongest response to hydrocarbons. Can it be connected?
- The preadsorbed oxygen (reactive oxygen) is needed for the formation of formates on the materials surfaces (see reference 23 for more details). We added a discussion to the manuscript (see below).
- The paragraph between lines 252 and 260 must be rewritten – the DRIFT spectra show you only HCO2- groups, not ethylene or acetylene molecule adsorption. Thus, addressing the observed sensor response to pi-MO system only is incorrect. The oxygen should play role as well. Maybe thermo-programmed reduction of obtained materials or thermo-programmed desorption of probe molecules can give rise to the supported ideas on the origins of sensor effect.
- We fully agree that the formation of surface formates from hydrocarbons requires a source of oxygen atoms in the reaction path. This aspect was not covered at all in this manuscript, but was addressed in our previous publication (see reference 23). Nevertheless, we added an explanatory paragraph also in the current manuscript. The reviewer is right, TPD experiments can be helpful and will be taken into consideration for further investigations.
Based on the currently available experimental results there is no proof of the role of π-orbitals in the response, but we did not attribute the sensing only to that. We will keep this hypothesis in the manuscript with the observation that there is no experimental proof for it yet.
- Lines 256-258 “In contrast to methane and ethane, acetylene and ethylene have π-MOs to interact with the metals. Their interaction with Fe-centres, where the dx2-y2 MO is occupied may be the key to the gas sensing mechanism [46].” This notion is vaguely supported by the provided reference. The authors should provide extended explanations.
- The reference was poorly placed and the whole paragraph was rephrased to distinguish literature reports from our own ideas more clearly. The section in the manuscript was changed to the following: “The molecular orbitals (MO) with symmetry eg centered at the B-site of the perov-skite with the highest energy and eg symmetry are populated by 0 (Cr3+), 1 (Mn3+, Co3+, Ni3+), or 2 (Fe3+) electrons and interact differently with simple gas molecules such as CO, NO or O2 [46]. The reaction sequence involves multiple intermediates, whose energies are dependent on the composition of the perovskite [47] The creation of formates requires preadsorbed oxygen, which is eventually provided by the oxygen in the gas phase. This is in line with the finding, that the materials with higher signals to oxygen also show a better response to hydrocarbons. Moreover, we assume that the reception of the gas molecules investigated in this study relies on the transfer of electrons between molecular orbitals of the hydrocarbons and the B-site cations of the perovskite. The structure of the analyte gases in this study is much more complicated than O2 or CO. In contrast to methane and ethane, acetylene and ethylene have additional π-MOs to interact with the metals. A possible explanation for the high sensor signals of ethylene and acetylene may be the interaction of these π-MOs with the dx2-y2 MO of the Fe centres. This orbital is occupied only for LnFeO3 [46]. This effect in combination with different reactivity to oxygen is suspected to be the cause of the different sensing performances of materials with different B-sites. The exchange of La with Sm or Gd does not change the number of electrons in the MOs at the B-site. However other factors such as ionic radii, electronegativity or magnetic properties might be the cause for the different sensor responses. Further research on the surface chemistry and DFT calculations are planned to investigate the interplay of the perovskite surface, oxygen, and the hydrocarbon molecules.”
- Lines 72-73 “We demonstrated that the sensor response of LFO is associated with the 72 formation of surface formates, not the formation/cancellation of surface vacancies.” – there are no investigation methods, reported in the manuscript, which deal with the oxygen vacancies. The sentence is unsupported.
- The sentence is supported by reference 23, which is one of our previous works. If the formation or cancellation of oxygen vacancies would play the determining role, the full combustion – which will produce the largest concentration of oxygen vacancies – will give the highest sensor signals. We demonstrated that the appearance of CO2, which indicates the full combustion, is not correlated to the largest sensor signal.
- Lines 79-80 “Moreover, operando DRIFT Spectroscopy was used to investigate the sensing 79 ” – talking about a molecular pathway of the reaction would be more correct, than using the term “sensing mechanism”.
Round 2
Reviewer 2 Report
The authors have definitely improved the manuscript so it can accepted to publication after minor corrections:
Comments on authors response:
1) * The materials effective surface area must be measured.
-
- We respectively disagree with the reviewer because the SEM results don’t indicate major differences in porosity and the differences in sensor signals between the materials cannot be attributed to different specific surfaces. Anyways, it will take us a lot of time to get these results because this technique is not available in house. Accordingly we are asking the reviewer to abandon his demand for this investigation.
Comment: materials on fig. 2 e,f,g demonstrate more or less developed mesoporous structure, contrary to what is observed in the case of a,b,c,d, where only macropores are present. I aggree that it can not govern the 3 orders of magnitude difference in response value between Fe-containing materials and others, however, it may expalin the difference between the La, Sm and Gd orthoferrites. I suppose that the authors can include this suggestion in the text.
- We respectively disagree with the reviewer because the SEM results don’t indicate major differences in porosity and the differences in sensor signals between the materials cannot be attributed to different specific surfaces. Anyways, it will take us a lot of time to get these results because this technique is not available in house. Accordingly we are asking the reviewer to abandon his demand for this investigation.
- 2) Lines 72-73 “We demonstrated that the sensor response of LFO is associated with the 72 formation of surface formates, not the formation/cancellation of surface vacancies.” – there are no investigation methods, reported in the manuscript, which deal with the oxygen vacancies. The sentence is unsupported.
- The sentence is supported by reference 23, which is one of our previous works. If the formation or cancellation of oxygen vacancies would play the determining role, the full combustion – which will produce the largest concentration of oxygen vacancies – will give the highest sensor signals. We demonstrated that the appearance of CO2, which indicates the full combustion, is not correlated to the largest sensor signal.
Comment: I have studied the reference 23 and found that the conclusion on involvement or absence of involvement of oxygen vacancies in the sensor response formation is speculative in a great extent. However, I hope that I've catched the idea of the authors and may offer the correction - it will be more correct to use term "oxidation-reduction of oxide surface" in this sentece, as it includes not only reactions with lattice oxygen, but the chemisorbed one as well, which is definitely playing its role in acetylene and ethylene sensing.
The other replies of authors are fully accepted.
Author Response
We agree with the suggestions of the reviewer and we changed the manuscript accordingly. We would like to thank him again for helping us to improve the manuscript.